# Sex differences in growth and mortality in pregnancy-associated hypertension

**Jess C. Hercus**[1], **Katherine X. Metcalfe**[1], **Julian K. Christians** [1,2,3,4]*

**1** Department of Biological Sciences, Simon Fraser University, Burnaby, British Columbia, Canada, **2** Centre for Cell Biology, Development and Disease, Simon Fraser University, Burnaby, BC, Canada, **3** British Columbia Children's Hospital Research Institute, Vancouver, BC, Canada, **4** Women's Health Research Institute, BC Women's Hospital and Health Centre, Vancouver, British Columbia, Canada

* julian_christians@sfu.ca

## Abstract

### Background

It is hypothesized that male fetuses prioritize growth, resulting in increased mortality, whereas females reduce growth in the presence of adversity. Preeclampsia reflects a chronic condition, in which fetuses have the opportunity to adjust growth. If females reduce their growth in response to preeclampsia, but males attempt to maintain growth at the cost of survival, we predict that differences in birthweight between preeclamptic and non-preeclamptic pregnancies will be greater among females, whereas differences in mortality will be greater among males.

### Methods

We analysed data from the Centers for Disease Control and Prevention. We compared pregnancies with pregnancy-associated hypertension (PAH) and controls.

### Results

The difference in birthweight between pregnancies affected by PAH and controls varied by fetal sex and gestational age. Among pregnancies of White individuals, at 34–35 weeks, the difference between PAH and controls was higher among females, as predicted. However, this pattern was reversed earlier in pregnancy and around term. Such variation was not significant in Black pregnancies. In both Black and White pregnancies, early in gestation, males had lower odds of death in PAH pregnancies, but higher odds of death in control pregnancies, counter to our prediction. Later, males had higher odds of death in PAH and controls, although the increased odds of death in males was not higher in PAH pregnancies than in controls. Overall, the difference in birthweight between surviving and non-surviving infants was greater in males than in females, opposite to our prediction.

### Conclusions

The impact of PAH on birthweight and survival varies widely throughout gestation. Differences in birthweight and survival between male and female PAH and controls are generally

**Data Availability Statement:** The datasets generated in this study are available in the National Center for Health Statistics Vital Statistics Online Data Portal, https://www.cdc.gov/nchs/data_access/vitalstatsonline.htm.

**Funding:** This study was funded by a Natural Sciences and Engineering Research Council of Canada Discovery Grant (JKC; grant number RGPIN-2021-02853). The funders had no role in study design, data collection and analysis, decision to publish, or preparation of the manuscript.

**Competing interests:** The authors have declared that no competing interests exist.

not consistent with the hypothesis that males prioritize fetal growth more than females, and that this is a cause of increased mortality in males.

## Introduction

Male infants are, on average, heavier at birth than female infants [1], but also have higher risks of perinatal mortality and other early adverse events [2,3]. Males are more likely to be born prematurely, particularly at earlier gestational ages [4]. Male infants are at higher risk of distress and asphyxia during labour, and have lower 5-minute Apgar scores [5]. Furthermore, males experience higher risks of various forms of pulmonary morbidity including respiratory distress syndrome, and increased need for oxygen and ventilator support [2]. Many authors have suggested that these differences may be due to males prioritizing growth at the expense of placental development, causing a higher risk of undernutrition if the pregnant person experiences adverse conditions, and that females are more adaptable to early-life challenges than males [6–8].

Sex differences in physiological traits in healthy fetuses suggest potential differences in prenatal growth strategy. Males and females have significantly different cardiovascular function as early as 30 weeks gestation [9]. Changes in brain connectivity during gestation also differ between the sexes, with females showing more development of long-range connections and males having more growth in local connectivity [10]. Males and females show differential expression of genes involved in several different functions in the placenta, providing a potential mechanism by which differential growth strategies could be achieved [11–19]. These could, in turn, contribute to sex differences in fetal oxygen extraction and maternal glucose metabolism [20,21].

The growth of males and females may respond differently to prenatal environment, although responses vary widely depending on the factor and timing of exposure. Higher maternal body mass index (BMI) is associated with higher birthweight in both sexes, but premature males are more affected than premature females, whereas at term the reverse is true [22]. Females born to asthmatic individuals who do not use glucocorticoids are significantly smaller compared to control females and females born to individuals using glucocorticoids, whereas males do not show any significant difference between groups [23]. Daily vigorous exercise significantly increases the chances of having a small for gestational age infant, but the risk is nearly twice as high for male infants compared to females [24].

Preeclampsia provides another model to examine whether the sexes respond differently to prenatal conditions. Preeclampsia is a major form of pregnancy-associated hypertension, affecting 2–8% of pregnancies globally [25–27]. Preeclampsia is defined by hypertension arising after 20 weeks gestation, in combination with other maternal symptoms, or with observed intrauterine growth restriction (IUGR) [28]. While preeclampsia is a heterogeneous syndrome, it is believed that most cases are caused by impaired placentation very early in pregnancy [29,30], leading to altered blood flow to the placenta throughout pregnancy and reduced nutrient and oxygen availability for the fetus. Due to this etiology, preeclampsia is a known factor in IUGR, often leading to small-for-gestational age (SGA) infants [26,31–33]. As preeclampsia is a chronic adverse condition *in utero*, we may expect to observe different growth strategies between sexes in response to this challenge [34] in addition to differing mortality rates.

The growth strategy hypothesis posits that males prioritize growth, resulting in increased mortality, whereas females are more adaptable, reducing growth in the presence of adversity,

and thereby reducing mortality [35]. However, this hypothesis has rarely been tested directly, and has not been examined using birthweight data in the context of preeclampsia, which is the goal of the present study. The premise of this work is that pregnancy-associated hypertension (PAH) will generally indicate a chronic condition, whereby fetuses will have had opportunity to adjust their growth in response to altered nutrient supply and *in utero* environment. In contrast, pregnancies without PAH or other potentially chronic complications will be more likely to reflect a fetus on a normal growth trajectory, even if born premature. Given this premise and the hypothesis that males prioritize growth whereas females reduce growth in the presence of adversity, we predict:

1. among surviving fetuses, differences in birthweight between PAH and non-PAH pregnancies will be greater among females than among males; females with PAH will have reduced growth whereas males will have attempted to maintain growth;

2. among all fetuses, differences in birthweight between fetuses that survive and those that die prenatally or shortly after birth will be greater among females than among males [36]; females facing serious prenatal insults will have reduced growth more than males;

3. the difference in mortality between the sexes will be higher among pregnancies affected by PAH; males will have attempted to maintain growth despite chronic adversity, resulting in reduced survival.

In addition, we also examined the effects of PAH on the odds of being born in males and females. We tested these predictions using data from the National Center for Health Statistics of the Centers for Disease Control and Prevention.

## Materials and methods

We used the public-use Birth Cohort Linked Birth–Infant Death Data Files from the National Center for Health Statistics of the Centers for Disease Control and Prevention [37]. We used data from years 1989–1991, and 1995–2003. Prior to 1989 and after 2005, data on some pregnancy complications are not available. Changes to the coding of race after 2003 had a substantial effect on sample sizes in some states. Linked data files are not available for 1992–1994.

To obtain a relatively homogeneous sample and avoid some potential confounds, we restricted analyses to singleton pregnancies with no record of a previous live birth and where offspring sex was assigned male or female, and to Non-Hispanic White or Black individuals between the ages of 18 and 34, inclusive; White or Black individuals were analysed separately. We placed restrictions on maternal age to avoid confounding effects of increased complications at young and advanced ages, e.g., increased rates of stillbirth, fetal growth restriction, small-for-gestational age, preterm birth at advanced maternal age [38]. We excluded pregnancies where the gestational age was below 24 weeks or over 42 weeks; gestational age was primarily based on the last menstrual period, but a clinical estimate was used if the date of the last menstrual period was not available or the gestational age was not consistent with birthweight. Imputed data were considered to be missing.

A pregnancy was coded as having pregnancy-associated hypertension (PAH) if there was a record of PAH or eclampsia, regardless of the presence of other complications. PAH was defined by an increase in blood pressure of at least 30 mm Hg systolic or 15 mm Hg diastolic on two measurements taken 6 hours apart after the 20th week of gestation. Eclampsia was defined as the occurrence of convulsions and/or coma unrelated to other cerebral conditions in women with signs and symptoms of pre-eclampsia. No information regarding severity of hypertension, or the presence of proteinuria was available in this dataset. A pregnancy was

coded as a control if there was no report of chronic hypertension, premature rupture of membranes, abruption or incompetent cervix. Pregnancies with one of these other complications, but not PAH or eclampsia, were excluded. We excluded data from states where more than 5% of observations had missing values for pregnancy complications (Connecticut, Hawaii, Kentucky, Maryland, Minnesota, Nevada, North Dakota, Oklahoma, Pennsylvania, Rhode Island, Texas, Vermont, Virginia and Washington), and from states where the sample sizes were less than 100 (Kentucky, Nevada, North Dakota, and Wyoming).

We excluded observations with implausible birthweights for gestational age by excluding those below the $0.5^{th}$ percentile or above the $99.5^{th}$ percentile by gestational age and sex according to Aris et al 2019 [39]. We used SAS (Version 9.4) for all analyses. Models are described in further detail below. Fig 1 provides a flow chart showing inclusions and exclusions and samples used in each analysis; the number of exclusions based on birthweight by gestational age are provided in S1 Table. Table 1 provides descriptive data for PAH and control pregnancies by sex, including covariates used in analyses.

## Results

### Prediction 1: Among surviving fetuses, differences in birthweight between PAH and controls will be greater among females than among males

We restricted this analysis to infants who were alive at 29 days after birth, to exclude neonatal deaths. We analysed birthweight with a general linear model (proc GLM) testing effects of gestational age as a categorical variable (to allow a non-linear relationship with birthweight), sex, group (PAH or control), tobacco use (yes or no), year and all possible pairwise and three-way interactions between gestational age, sex and group (S2 Table). Overall, there was a significant effect of both sex and group on birthweight, with males heavier than females and controls heavier than PAH (S1 Fig). With regards to our prediction that differences in birthweight between PAH and controls would be greater among females, there was no significant interaction between sex and group in White or Black individuals (S2 Table). However, among White individuals, there was a three-way interaction between gestational age, sex and group (S2 Table), indicating that the strength of the sex by group interaction varied with gestational age. At 34–35 weeks, the difference between PAH and controls was higher among females, as predicted (Fig 2). However, this pattern was reversed at 29–30 weeks and around term (Fig 2). There was also a significant interaction between group and gestational age (S2 Table), such that the difference between groups was greatly reduced around term (Figs 2 and S1). The three-way interaction between gestational age, sex and group was not significant among Black individuals (S2 Table), although patterns were similar (Fig 2).

### Prediction 2: Among all fetuses, differences in birthweight between fetuses that survive and those that die prenatally or shortly after birth will be greater among females than among males

In this analysis, we included fetal deaths and infants who died within 28 days of birth as non-survivors. We repeated the analysis above, replacing group (PAH or control) with survival (yes or no) in interaction terms, but retaining the main effect of group, i.e., we analysed birthweight testing effects of gestational age, sex, survival, group, tobacco use, year and all possible pairwise and three-way interactions between gestational age, sex and survival (S3 Table). There was a significant effect of survival on birthweight, with surviving infants being heavier (S2 Fig). There was a significant interaction between sex and survival (S3 Table) whereby the difference in birthweight between surviving and non-surviving infants was greater in males than in

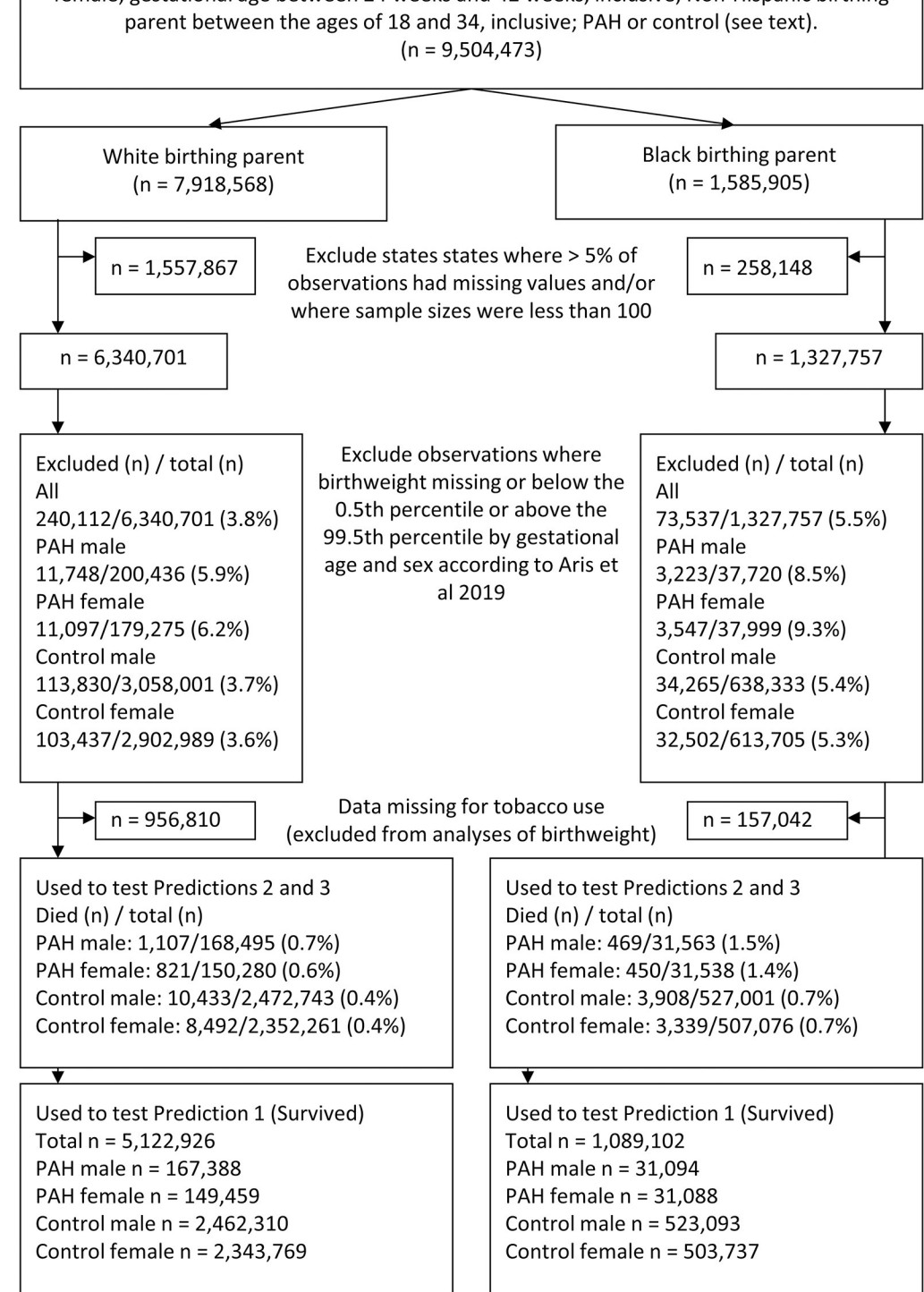

**Fig 1. Flow chart showing inclusion and exclusion criteria, and sample sizes for each analysis.**

**Table 1. Descriptive statistics for pregnancy-associated hypertension (PAH) and control pregnancies.** Percentages refer to percent of total for a given subgroup (e.g., for White PAH males).

| | White birthing parent | | | | Black birthing parent | | | |
|---|---|---|---|---|---|---|---|---|
| | **PAH male** | **PAH female** | **Control male** | **Control female** | **PAH male** | **PAH female** | **Control male** | **Control female** |
| N | 168495 | 150280 | 2472743 | 2352261 | 31563 | 31538 | 527001 | 507076 |
| Birthweight (g) Mean ± SD | 3210 ± 672 | 3095 ± 668 | 3433 ± 505 | 3320 ± 473 | 2940 ± 749 | 2782 ± 762 | 3215 ± 549 | 3105 ± 526 |
| Gestational age (weeks) Mean ± SD | 38.1 ± 2.5 | 38.1 ± 2.6 | 39.2 ± 1.9 | 39.3 ± 1.8 | 37.5 ± 3.0 | 37.3 ± 3.3 | 38.7 ± 2.4 | 38.7 ± 2.3 |
| Term (37 weeks or greater) | 138484 (82.2%) | 124014 (82.5%) | 2299166 (93.0%) | 2211598 (94.0%) | 23251 (73.7%) | 22439 (71.2%) | 467737 (88.8%) | 451822 (89.1%) |
| Moderately preterm (32–36 weeks) | 25677 (15.2%) | 21927 (14.6%) | 156450 (6.3%) | 127726 (5.4%) | 6586 (20.9%) | 6849 (21.7%) | 49579 (9.4%) | 46287 (9.1%) |
| Very preterm (28–31 weeks) | 3442 (2.0%) | 3277 (2.2%) | 11112 (0.5%) | 8157 (0.4%) | 1287 (4.1%) | 1604 (5.1%) | 5413 (1.0%) | 5100 (1.0%) |
| Extremely preterm (less than 28 weeks) | 892 (0.5%) | 1062 (0.7%) | 6015 (0.2%) | 4780 (0.2%) | 439 (1.4%) | 646 (2.1%) | 4272 (0.8%) | 3867 (0.8%) |
| Mortality | 1107 (0.66%) | 821 (0.55%) | 10433 (0.42%) | 8492 (0.36%) | 469 (1.47%) | 450 (1.43%) | 3908 (0.74%) | 3339 (0.66%) |
| Tobacco use | 19362 (11.5%) | 17180 (11.4%) | 363722 (14.7%) | 343587 (14.6%) | 1563 (5.0%) | 1469 (4.7%) | 29586 (5.6%) | 27987 (5.5%) |

females, opposite to our prediction (Fig 3). While the interaction was statistically significant, the magnitude of the difference in birthweight between surviving and non-surviving infants was similar in males and females in both White (males: 206 ± 4 g; females: 183 ± 5 g) and Black (males: 207 ± 7 g; females: 161 ± 8 g) pregnancies. The magnitude of these differences fluctuated with gestational age (Fig 3), but the three-way interaction between sex, survival and gestational age was not significant (S3 Table), indicating that the interaction between sex and survival did not vary with gestational age.

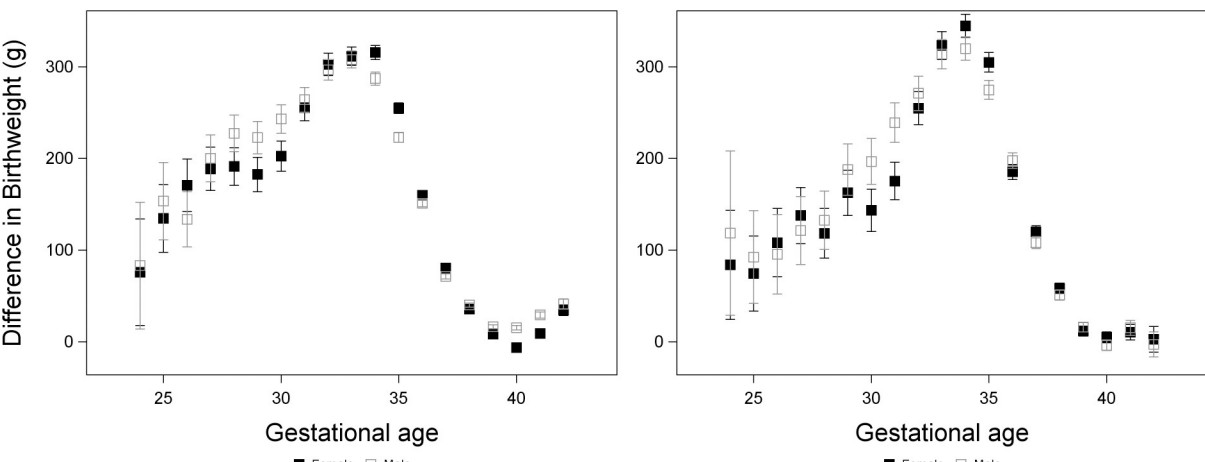

**Fig 2.** Differences in birthweight between PAH pregnancies and controls among surviving fetuses in White pregnancies (left) and Black pregnancies (right). Differences ± standard errors were calculated using the ESTIMATE statement of proc GLM and were from a model including gestational age as a categorical variable, sex, group (PAH or control), tobacco use (yes or no), year and all possible pairwise and three-way interactions between gestational age, sex and group.

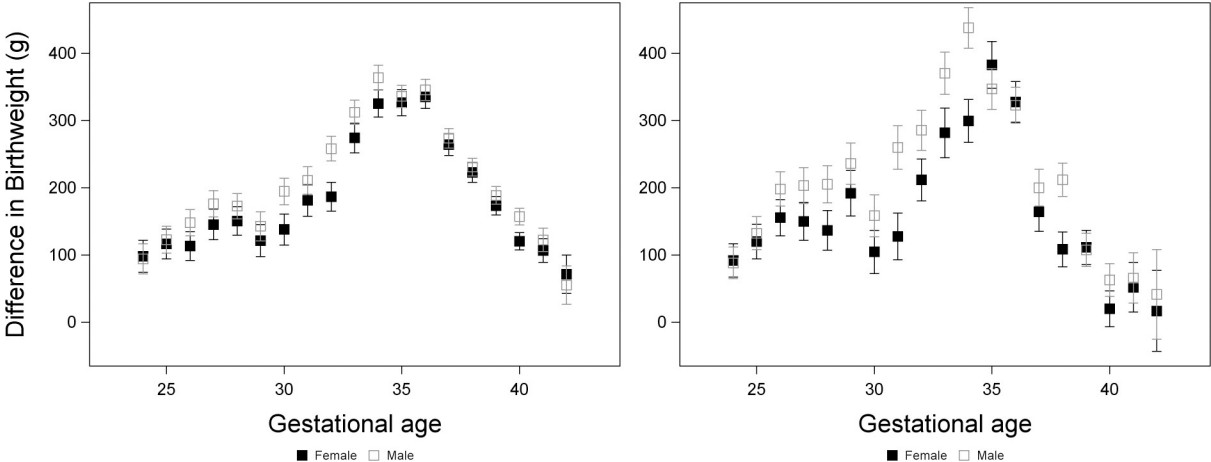

**Fig 3.** Differences in birthweight between infants that died prenatally or within 28 days and surviving infants in White pregnancies (left) and Black pregnancies (right). Differences ± standard errors were calculated using the ESTIMATE statement of proc GLM and were from a model including gestational age as a categorical variable, sex, survival, group, tobacco use (yes or no), year and all possible pairwise and three-way interactions between gestational age, sex and survival.

## Prediction 3: The difference in mortality between the sexes will be higher among pregnancies affected by PAH

We first examined mortality using a fetus-at-risk approach [40,41], testing survival time using a proportional hazards model (proc PHREG), including effects of sex, group (PAH or control) and the interaction between sex and group. In this analysis, we calculated time of death from the beginning of gestation (e.g., an infant born at 24 weeks that died 5 weeks after birth would be considered to have died at 29 weeks), and censored surviving infants at 46 weeks. However, we found the proportional hazards assumption to be violated (ZPH option; P < 0.001 for all terms in the model); the effects of sex, group and the interaction between sex and group all varied with time since the beginning of gestation (Fig 4).

We therefore performed a logistic regression (proc LOGISTIC) analysis to examine the effects of sex, group and the interaction between sex and group on the odds of dying at specific time points. This analysis used a sliding window where, at each week, we analysed the odds of

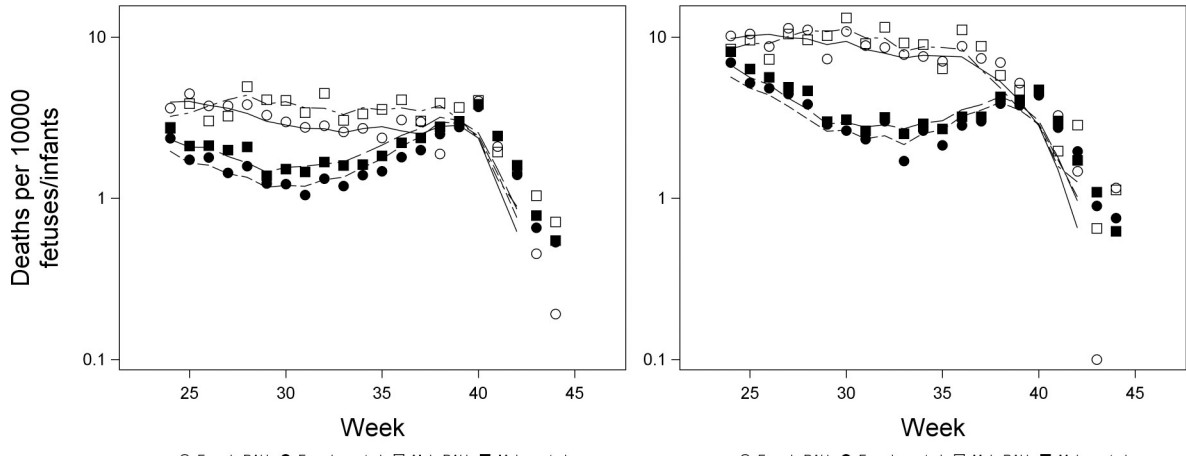

**Fig 4.** Effects of sex and PAH on mortality using a fetus-at-risk approach in White pregnancies (left) and Black pregnancies (right). Symbols denote deaths at a single week, whereas lines are calculated using a sliding window that averages the deaths at a focal week and in the following two weeks.

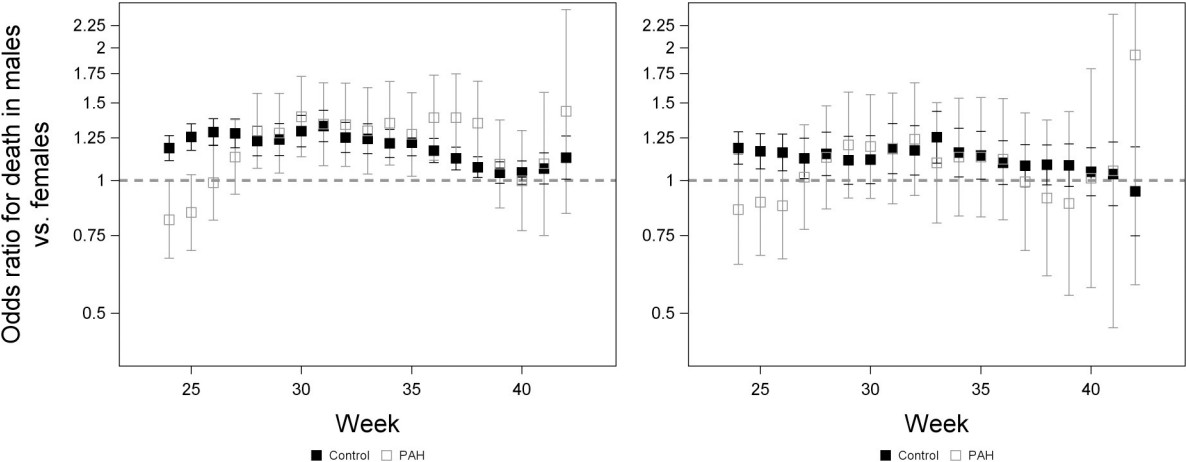

**Fig 5.** The odds of death for males relative to females in PAH pregnancies and controls in White pregnancies (left) and Black pregnancies (right). An OR > 1 indicates that males are at greater risk of death than females within a group. A sliding window was used to analyse the odds of death at that week or in the following two weeks, including fetal deaths and deaths within the first 28 days after birth, and including all fetuses at risk. Odds ratios for the effect of sex were calculated using the ODDSRATIO statement of proc LOGISTIC and were from a model including effects of sex, group and the interaction between sex and group. Error bars denote 95% confidence limits.

death at that week or in the following two weeks, including fetal deaths and deaths within the first 28 days after birth. This analysis included all fetuses at risk, i.e., all those still *in utero* or less than 28 days old. Among White pregnancies, the interaction between sex and group was significant early in gestation (weeks 24–26) with males having lower odds of death in PAH pregnancies, but higher odds of death in control pregnancies, counter to our prediction (Fig 5; S4 Table); in Black pregnancies, this interaction was significant in week 24, but marginally non-significant in weeks 25 and 26 (P = 0.08 and P = 0.06, respectively; S5 Table). From weeks 27 to 38, males in White pregnancies had higher odds of death in both groups, although for most of this time, increased risk did not differ between groups (the sex by group interaction was not significant) (Fig 5; S4 Table). In week 38, the interaction between sex and group was significant (P = 0.045), with the increased odds of death in males being higher in PAH pregnancies, as predicted (Fig 5; S4 Table). In Black pregnancies, the odds of death differed between males and females only in weeks 31 and 32 (S5 Table). Around term, there was no difference in the odds of death between males and females. From 24–37 weeks, there was significantly higher mortality in PAH pregnancies than in controls in both White and Black pregnancies (P < 0.02 in all weeks; Fig 4; S4 and S5 Tables).

## Sex differences in the odds of birth in preeclampsia and controls

It is known that males are more likely to be born premature [4,42–44]. However, we examined whether this increased risk of preterm birth in males differed between PAH pregnancies and controls. We performed a logistic regression (proc LOGISTIC) analysis to examine the effects of sex, group and the interaction between sex and group on the odds of being born at specific gestational ages. As above, this analysis used a sliding window where, at each week, we analysed the odds of birth at a given week or in the following two weeks. Only live births were considered, although the fetuses at risk included fetal deaths that occurred later in gestation.

Early in gestation, control males had higher odds of being born than control females, whereas PAH females had higher odds of being born than PAH males (Figs 6 and 7). In White pregnancies, in weeks 34–38, males had higher odds of being born than females in both groups, but this sex difference was greater in controls (Fig 7). The interaction between sex and

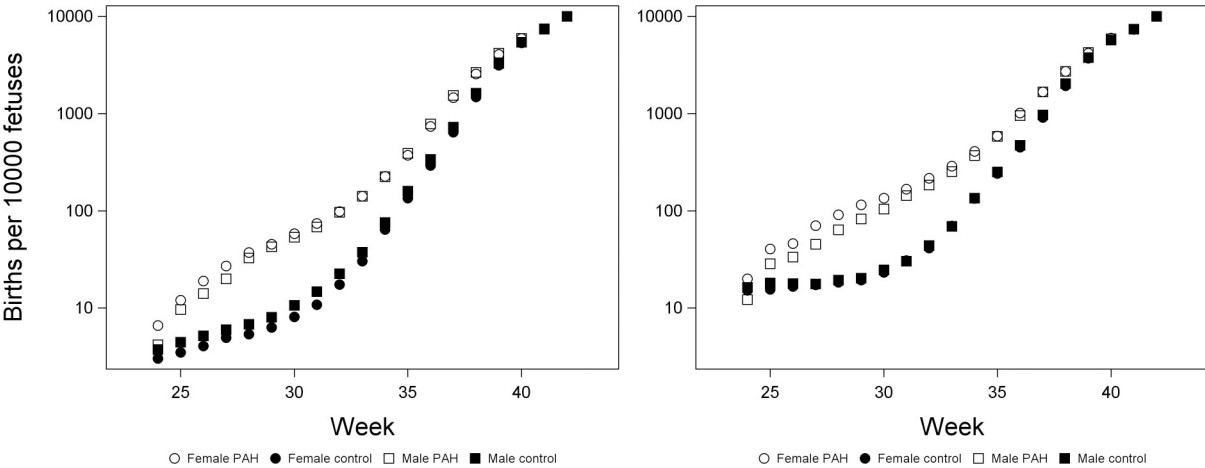

**Fig 6.** Effects of sex and PAH on births using a fetus-at-risk approach in White pregnancies (left) and Black pregnancies (right). Because we excluded pregnancies where the gestational age was over 42 weeks, all remaining "fetuses-at-risk" were born at 42 weeks, i.e., 10000 births per 10000 fetuses.

group was significant in weeks 24–37 in Black pregnancies and weeks 24–38 in White pregnancies. Throughout gestation, the odds of being born were higher in PAH pregnancies than in controls (Fig 6).

## Discussion

### Prediction 1: Among surviving fetuses, differences in birthweight between PAH and controls will be greater among females than among males

Birthweights were lower in PAH pregnancies than in controls, likely reflecting the increased prevalence of a chronic condition in the PAH group. However, this difference was not consistently greater in one sex than the other. Overall, the interaction between sex and group was not significant, i.e., the difference in birthweight between PAH and controls did not differ

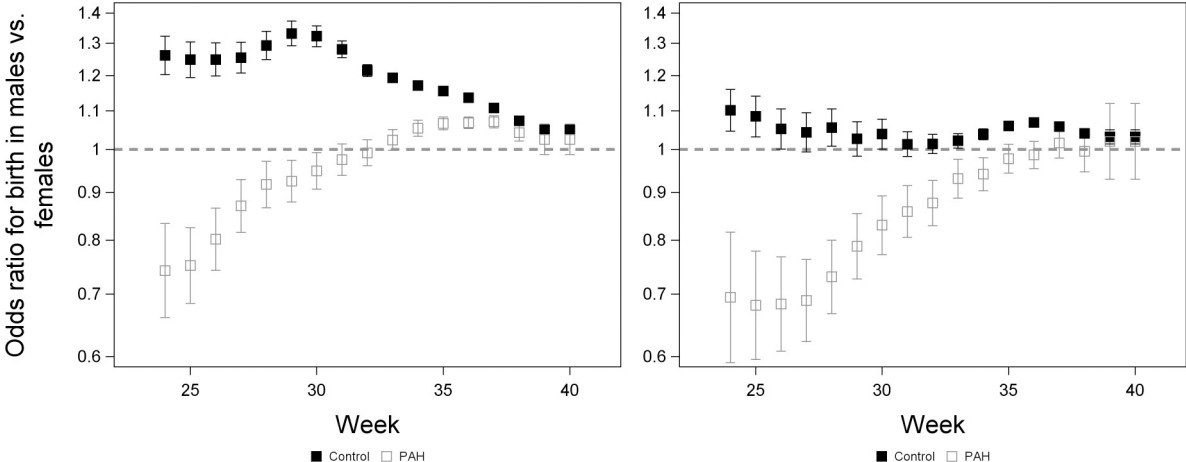

**Fig 7.** The odds of birth for males relative to females in PAH pregnancies and controls in White pregnancies (left) and Black pregnancies (right). An OR > 1 indicates that odds of birth are greater for males than females within a group. A sliding window was used to analyse the odds of birth at that week or in the following two weeks, including all fetuses at risk. Odds ratios for the effect of sex were calculated using the ODDSRATIO statement of proc LOGISTIC and were from a model including effects of sex, group and the interaction between sex and group. Error bars denote 95% confidence limits.

between sexes. However, among White pregnancies, this interaction did vary significantly across gestational ages, with males showing a larger difference between PAH and controls compared to females around 29–30 weeks, and again around 40–41 weeks, but with females showing the reverse pattern around 34–35 weeks. Black pregnancies showed similar patterns, although not statistically significant, perhaps due to a lower sample size. A previous study found that preterm males reduced their growth more in response to preeclampsia, which agrees with our results [34]. While not consistent with our prediction, this result might indicate that pregnancies with male fetuses are associated with more severe forms of PAH/ preeclampsia, although we did not have information regarding the severity of the condition. We found that the difference in birthweight between PAH and control pregnancies was reduced around term, which is consistent with late-onset preeclampsia being a less severe condition than early-onset preeclampsia [45].

### Prediction 2: Among all fetuses, differences in birthweight between fetuses that survive and those that die prenatally or shortly after birth will be greater among females than among males

Based on the growth strategy hypothesis, we would expect females to reduce their growth in response to prenatal insults, whereas males should sustain their growth [35]. This should lead to a larger difference in birthweight between females who survived and those who died due to prenatal insults. However, we found the opposite: throughout gestation, males had a significantly greater difference between the birthweight of surviving and non-surviving fetuses compared to females. This contrasts with our previous work in a different cohort, where we found that prior to 33 weeks, the difference in birthweight between surviving and non-surviving fetuses was greater in females [36]. However, in this previous work, we had not restricted observations based on maternal race or age, or the presence of specific pregnancy complications. The additional restrictions in the present study may have reduced confounding effects (e.g., differences in the prevalence of complications) and provided more accurate patterns. Alternatively, births in the previous cohort [36] occurred from 1959 to 1966 whereas those in the present study occurred from 1989 to 2003, and so changes in clinical management may have led to real changes in patterns of birthweight and survival.

### Prediction 3: The difference in mortality between the sexes will be higher among pregnancies affected by PAH

At 24–37 weeks of gestation, PAH pregnancies had significantly higher mortality compared to control pregnancies. This is consistent with other studies that have found infants exposed to PAH have higher risks of stillbirth and neonatal mortality [46,47]. Early in gestation, males had a higher risk of death in control pregnancies than females, but a lower risk in PAH pregnancies, inconsistent with our prediction. This reversal of sex differences in PAH pregnancies may indicate that females are exposed to more severe PAH at very early gestational ages (weeks 24–26). From weeks 27–38, males were more likely to die than females in both PAH and control groups in White pregnancies, which is consistent with other studies finding higher stillbirth risks for males [3,48]. However, sex did not moderate the effect of PAH on mortality risk, i.e., males exposed to PAH were not disproportionately more likely to die.

### Sex differences in the odds of birth in preeclampsia and controls

Early in gestation, control males had higher odds of being born than control females, whereas the opposite was true among PAH pregnancies. Later in pregnancy, males had a higher odds

of being born than females in both groups in White pregnancies, but this sex difference was greater in controls. These results are consistent with previous findings that overall males are at greater risk of premature birth [4,42–44], although we found this male bias to be reduced among Black pregnancies. While our analysis did not examine the rates of PAH in males and females (i.e., proportion of PAH pregnancies among all pregnancies), the female bias in births early in gestation among PAH pregnancies may reflect females being exposed to a more severe condition, and/ or being exposed earlier in gestation; female pregnancies are at increased risk of preterm preeclampsia [49,50].

## Limitations

Our study only included non-Hispanic White and Black primiparous individuals from 18–34 years of age, leading to more homogenous samples. However, our results may not be generalizable to other groups. We were not able to distinguish between preterm and term preeclampsia, and we could not determine the severity of preeclampsia, which could be confounding if one sex is more likely to experience a more severe form of the disorder. Indeed, female pregnancies are at increased risk of preterm preeclampsia, whereas male pregnancies are at increased risk of term preeclampsia [49,50]. However, a greater incidence of early, severe preeclampsia among females would be expected to cause a greater difference in birthweight between PAH and controls among females than among males, which was not observed prior to 34 weeks. The prevalence of various causes of preterm birth among controls may have also differed between the sexes. The lack of clinical information also precluded identification of potential mechanisms underlying reduced fetal growth. For example, during healthy pregnancy, there is an increase in maternal cardiac output and blood volume, along with reduced systemic vascular resistance and blood pressure [51]. Failure to undergo these normal changes is associated with pre-eclampsia (PE) and fetal growth restriction [52–55], perhaps because of impairment of placental development and growth [55]. Inconsistent data collection could have introduced bias into the sample, although it is not clear how such biases would have affected one sex more than the other. While we did exclude states if more than 5% of observations had missing values, there may be undetected bias if, for example, rural or underserved areas were more likely to have incorrectly recorded values.

## Conclusions

Overall, results regarding sex differences were similar among White and Black pregnancies, despite the rates of hypertensive disorders of pregnancy, SGA and preterm birth differing between these groups [56–58]. Among surviving infants, the impact of PAH on birthweight varies throughout gestation; early in gestation, males reduce their birthweight more, while this reverses around late pre-term so that females are more affected, with males again being more affected at term. This result is not consistent with the hypothesis that males and females have different growth strategies; if the sexes were employing different strategies in response to PAH, we would expect this to be consistent throughout gestation. Furthermore, non-surviving males show greater reductions in birthweight than non-surviving females, opposite to predictions based on the growth strategy hypothesis. Finally, males are not disproportionately more likely to die due to PAH exposure. While these results may reflect sex differences in the prevalence and severity of pregnancy complications, rather than growth strategies, they do not provide any support for the growth strategy hypothesis, indicating that other explanations for the excess mortality and morbidity experienced by male infants should be considered.

## Supporting information

**S1 Table.** Number of exclusions based on birthweight by gestational age in (A) White live births, (B) Black live births, (C) White fetal deaths, and (D) Black fetal deaths.
(XLSX)

**S2 Table. Effects on birthweight among infants who were still alive at 29 days after birth among White (N = 5122926) and Black (N = 1089012) individuals.**
(DOCX)

**S3 Table. Effects on birthweight, including fetal deaths and infants who died within 28 days of birth among White (N = 5143779) and Black (N = 1097178) individuals.**
(DOCX)

**S4 Table. The odds of death for males relative to females in PAH pregnancies and controls in White pregnancies.**
(DOCX)

**S5 Table. The odds of death for males relative to females in PAH pregnancies and controls in Black pregnancies.**
(DOCX)

**S1 Fig.** Effects of sex and PAH on birthweight in White pregnancies (left) and Black pregnancies (right).
(DOCX)

**S2 Fig.** Effects of sex and survival on birthweight in White pregnancies (left) and Black pregnancies (right).
(DOCX)

## Acknowledgments

We thank Bernard Crespi, K.S. Joseph, and Pablo Nepomnaschy for helpful feedback and discussion, and two anonymous reviewers for constructive comments.

## Author Contributions

**Conceptualization:** Jess C. Hercus, Katherine X. Metcalfe, Julian K. Christians.

**Data curation:** Jess C. Hercus, Katherine X. Metcalfe, Julian K. Christians.

**Formal analysis:** Julian K. Christians.

**Funding acquisition:** Julian K. Christians.

**Writing – original draft:** Jess C. Hercus, Julian K. Christians.

**Writing – review & editing:** Katherine X. Metcalfe.

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
