## [Decision Letter · Decision Letter 0]

7 Jun 2023

PONE-D-23-03980Sex differences in growth and mortality in pregnancy-associated hypertensionPLOS ONE

Dear Dr. Christians,

Thank you for submitting your manuscript to PLOS ONE. After careful consideration, we feel that it has merit but does not fully meet PLOS ONE’s publication criteria as it currently stands. Therefore, we invite you to submit a revised version of the manuscript that addresses the points raised during the review process.

We look forward to receiving your revised manuscript.

Kind regards,

Nur Aizati Athirah Daud, Ph.D.

Academic Editor

PLOS ONE

Journal Requirements:

Reviewers' comments:

Reviewer's Responses to Questions

**Comments to the Author**

1. Is the manuscript technically sound, and do the data support the conclusions?

Reviewer #1: Partly

Reviewer #2: Partly

2. Has the statistical analysis been performed appropriately and rigorously? 

Reviewer #1: I Don't Know

Reviewer #2: I Don't Know

3. Have the authors made all data underlying the findings in their manuscript fully available?

Reviewer #1: No

Reviewer #2: No

4. Is the manuscript presented in an intelligible fashion and written in standard English?

Reviewer #1: Yes

Reviewer #2: Yes

5. Review Comments to the Author

Reviewer #1: Thank you for inviting to review this manuscript for PLOS One.

The topic of the research is of epidemiological and physiology interest only, with no implications for the clinical practice irrespective of the study findings themselves. Furthermore, there are major weaknesses as below:

- The definition of hypertension is too vaque and not consistent with current definitions.

- another major limitation to the study findings is represented by tha acknowledged role of maternal haemodynamics in the patophysiology of hypertensive disorders of the pregnancy and on fetal growth. I cannot see one line on the role of maternal haemodynamics and hypertension in pregnancy despite the extensive research so far conducted. I understand this is not of interest for the Authors, but the paper needs such information to be added. Fetal gender is, of course, independent from maternal hemodynamics.

- I cannot see anywhere the number of the overall included cases and the numbers involved in sub-analysis. They all should be added to the text.

Additional comments

- Line 324: the concept of early- and late-preeclampsia is outdated

- Line 325: according to the latest definition of preeclampsia proposed by the Intenational Society for the Study of Hypertension in Pregnancy (ISSHP) preeclampsia is no longer distinguished into mild, moderate and severe.

Reviewer #2: Sex differences in growth and mortality in pregnancy-associated hypertension.

This manuscript puts together interesting empirical tests of the hypothesis that male fetuses prioritize growth in the presence of adversity in contrast to female fetuses who reduce growth. The authors compare male and female mortality and birthweight of fetuses in pregnancies with and without preeclampsia. The authors use public-use Birth Cohort Linked Birth – Infant Death Data Files from the National Center for Health Statistics of the Centers for Disease Control and Prevention from the years 1989-1991, and 1995-2003. While their way of thinking about this hypothesis is interesting, it is difficult to evaluate this manuscript because key pieces of information are not included and the population selection criteria may introduce bias.

The authors apply a large number of exclusion criteria for the definition of their population. Most are reasonable.They focus on White and non-Hispanic Black nulliparous women (reasonable) with a singleton pregnancy and valid sex. They also remove states with high numbers of missing data or small samples. However, they also set an age restriction which may not make sense since the outcome is associated with age - more justification is needed for this criterion. Furthermore, they exclude observations with implausible birthweights for gestational age, defined as those below the 0.5th percentile or above the 99.5th percentile by gestational age and sex. However, as their interest is in cases with potentially very low birthweight, including stillbirths with IUGR, more information on these exclusions are needed.

There was no descriptive table of the sample and I could not find the sample sizes for any of the analyses, so it was not clear how these restrictions affected the sample. In addition to this table, some analysis on the sex and preeclampsia of excluded fetuses is needed. The absence of these tables also prevented the evaluation of whether the data were plausible and how close they are to known incidence of the outcomes being evaluated.

Other selection decisions may also induce bias. The first analysis starts by looking at differences in birthweight, but only among births surviving to 29 days. This would exclude severe cases of preeclampsia and deaths related to IUGR. If there is differential mortality by sex (which is one of the hypotheses), this could affect this analysis.

In general, in the absence of descriptive tables, it was difficult to follow the analyses. The graphs were very difficult to read and interpret because they present a lot of information with overlapping confidence intervals and large scales. The multivariable models were not possible to interpret without baseline information.

I would suggest that the authors provide full descriptive tables of their exposure and outcomes and consider showing some of the data now in graphs as figures.

One issue to consider is that preterm births for preeclampsia and IUGR mainly occur if the fetus is detected as having a problem and is an indicated birth. This information is important for understanding these relationships because most fetal biometric charts used to interpret ultrasound measures are not sex-specific. Therefore, more female fetuses may be detected and also have resulting clinically decided preterm births. This information (prelabor cesarean) should be part of the information provided in descriptive tables.

6. PLOS authors have the option to publish the peer review history of their article (what does this mean?). If published, this will include your full peer review and any attached files.

Reviewer #1: No

Reviewer #2: No

---

## [Author Response · Author response to Decision Letter 0]

26 Jun 2023

We have uploaded a "Response to Reviewers" file.

---

## [Decision Letter · Decision Letter 1]

23 Oct 2023

PONE-D-23-03980R1Sex differences in growth and mortality in pregnancy-associated hypertensionPLOS ONE

Dear Dr. Christians,

Thank you for submitting your manuscript to PLOS ONE. After careful consideration, we feel that it has merit but does not fully meet PLOS ONE’s publication criteria as it currently stands. Therefore, we invite you to submit a revised version of the manuscript that addresses the points raised during the review process.

We look forward to receiving your revised manuscript.

Kind regards,

Nur Aizati Athirah Daud, Ph.D.

Academic Editor

PLOS ONE

Reviewers' comments:

Reviewer's Responses to Questions

**Comments to the Author**

1. If the authors have adequately addressed your comments raised in a previous round of review and you feel that this manuscript is now acceptable for publication, you may indicate that here to bypass the “Comments to the Author” section, enter your conflict of interest statement in the “Confidential to Editor” section, and submit your "Accept" recommendation.

Reviewer #2: (No Response)

Reviewer #3: (No Response)

2. Is the manuscript technically sound, and do the data support the conclusions?

Reviewer #2: Partly

Reviewer #3: No

3. Has the statistical analysis been performed appropriately and rigorously? 

Reviewer #2: Yes

Reviewer #3: No

4. Have the authors made all data underlying the findings in their manuscript fully available?

Reviewer #2: Yes

Reviewer #3: Yes

5. Is the manuscript presented in an intelligible fashion and written in standard English?

Reviewer #2: Yes

Reviewer #3: No

6. Review Comments to the Author

Reviewer #2: Thank you to the authors for their attention to the comments in the first round of reviews.

I remain concerned about the exclusions, which may impact on the results since they may exclude many growth restricted fetuses (which is a focus of the paper). The authors remove births with BW<.05 percentile. However, they use birthweight charts which are well known to be biased for preterm births (the charts are constructed from births, many of whom are growth restricted). This will be true in particular for births with fetal growth restriction. The authors should compare their exclusions, not just on the full population, but on their outcomes - the numbers excluded by gestational age for live and stillbirths by sex and by group. It is possible that there are many more exclusions in these high risk groups. The authors note that they have an equivalent proportion of exclusions at higher birthweights, but GA errors are much more likely to be in the direction of term babies being improperly coded as preterm (i.e. 38 as 28, given so many more 38 weekers, for instance), so having more high birthweights would be expected.

It would also be nice to have an introductory table with overall descriptive data for the two groups and sex, the distribution by GA, the covariables and mortality.

I am not sure why the authors use a sliding scale for their fetus at risk calculations - this makes it difficult to compare the results with other samples since this is not the usual practice.

Also, do the adjustments change the OR or differences in BW? It is not possible to know this from the data presented.

Finally, if the authors are worried about congenital anomalies and birthweight, wouldn't it make sense to remove those with congenital anomaly codes only? Also, this problem would also be present for the mortality analysis, right? If the authors want to do their analysis on survivors after 28 days, they might consider presenting the mortality analyses first since this respects the time sequence of events and may be less confusing.

Reviewer #3: 1) Method

Line 124 : Have the authors considered gestational diabetes as a confounding factor? GDM can affect the fetal

weight

2) Result

Line 229 : P=0.08 is non-signifcant, not marginally non-significant

The sentence is confusing 'marginally non-significant or significant'.. especially due to the choice of word

'or'. Please rephrase this sentence.

Line 232 : Stating the P value as P<0.1 is inadequate and cannot be regarded as significant. Please state the

actual P value, as the value from 0.05 - 0.1 is huge and this value determines the significance of the

finding.

3) Discussion

Line 289 - 290 : 'A previous study....' This statement agrees with the study's result. But the study's result was

contradictory to the hypothesis. Please discuss the reasons why the finding of the study was

inconsistent with the hypothesis.

Line 301 : 'However we found the opposite.' The author did not discuss the possible reasons as to why the

restrictions in the new cohort vs. non-restrictions in the previous cohort influenced the result.

Line 312 - 314 : Please discuss the possible reasons why higher deaths in control vs PAH pregnancies in early

gestation.. this is in order to address possible confounding factors that had caused this result

Line 316 - 318: This statement implies that sex does not influence mortality in PAH pregnancies. But the conclusion

in the Abstract (line 37 - 39) is contradictory to this statement (further details below).

4) Conclusion

Line 360 - 368 : These statements clearly stated that the results did not support the growth theory hypothesis. But the

Conclusion in the Abstract stated differently and is misleading to the whole content of the study.

7. PLOS authors have the option to publish the peer review history of their article (what does this mean?). If published, this will include your full peer review and any attached files.

Reviewer #2: No

Reviewer #3: No

---

## [Author Response · Author response to Decision Letter 1]

4 Dec 2023

Please see attachment uploaded with files.

---

## [Decision Letter · Decision Letter 2]

20 Dec 2023

Sex differences in growth and mortality in pregnancy-associated hypertension

PONE-D-23-03980R2

Dear Dr. Christians,

We’re pleased to inform you that your manuscript has been judged scientifically suitable for publication and will be formally accepted for publication once it meets all outstanding technical requirements.

Kind regards,

Nur Aizati Athirah Daud, Ph.D.

Academic Editor

PLOS ONE

Additional Editor Comments (optional):

Reviewers' comments:

Reviewer's Responses to Questions

**Comments to the Author**

1. If the authors have adequately addressed your comments raised in a previous round of review and you feel that this manuscript is now acceptable for publication, you may indicate that here to bypass the “Comments to the Author” section, enter your conflict of interest statement in the “Confidential to Editor” section, and submit your "Accept" recommendation.

Reviewer #2: All comments have been addressed

Reviewer #3: All comments have been addressed

2. Is the manuscript technically sound, and do the data support the conclusions?

Reviewer #2: (No Response)

Reviewer #3: Yes

3. Has the statistical analysis been performed appropriately and rigorously? 

Reviewer #2: (No Response)

Reviewer #3: Yes

4. Have the authors made all data underlying the findings in their manuscript fully available?

Reviewer #2: (No Response)

Reviewer #3: Yes

5. Is the manuscript presented in an intelligible fashion and written in standard English?

Reviewer #2: (No Response)

Reviewer #3: Yes

6. Review Comments to the Author

Reviewer #2: (No Response)

Reviewer #3: Feedback:

The author(s) has/have satisfactorily clarified/corrected the manuscript according to the feedback.

Only one comment:

Line 229-230 : ‘marginally non-significant’

Suggest the author(s) to remove the word ‘marginally’

7. PLOS authors have the option to publish the peer review history of their article (what does this mean?). If published, this will include your full peer review and any attached files.

Reviewer #2: No

Reviewer #3: No

---

## [Editor Report · Acceptance letter]

4 Jan 2024

PONE-D-23-03980R2 

PLOS ONE

Dear Dr. Christians, 

I'm pleased to inform you that your manuscript has been deemed suitable for publication in PLOS ONE. Congratulations! Your manuscript is now being handed over to our production team.

Kind regards, 

on behalf of

Dr. Nur Aizati Athirah Daud 

Academic Editor

PLOS ONE